# Enzymatic Birch reduction via hydrogen atom transfer at [4Fe-4S]-OH$_2$ and [8Fe-9S] clusters

Jonathan Fuchs [1], Unai Fernández-Arévalo [2], Ulrike Demmer[3], Eduardo Díaz[2], G. Matthias Ullmann [4], Antonio J. Pierik [5], Ulrich Ermler [3] & Matthias Boll [1] ✉

The alkali metal- and ammonia-dependent Birch reduction is the classical synthetic method for achieving dihydro additions to arenes, typically yielding 1,4-cyclodienes. A mild biological alternative to this process are 1,5-dienoyl-coenzyme A (CoA)-forming class I and II benzoyl-CoA reductases (BCRs), widely abundant key enzymes in the biodegradation of aromatic compounds at anoxic environments. To obtain a comprehensive mechanistic understanding of class I BCR catalysis, we produced the active site subunits from a denitrifying bacterium and determined the X-ray structure of its substrate and product complexes at 1.4 Å revealing non-canonical double-cubane [8Fe-9S] and active site aqua-[4Fe-4S] clusters. Together with kinetic, spectroscopic and QM/MM studies, we provide evidence for a radical mechanism with a [4Fe-4S] cluster-bound water molecule acting as hydrogen atom and electron donor at potentials beyond the biological redox window. An analogous Birch-like radical mechanism is applied by class II BCRs with the catalytic water bound to a tungsten-*bis*-metallopterin cofactor. The use of activated, metal-bound water ligands as hydrogen atom donor serves as a basic blueprint for future enzymatic or biomimetic Birch reduction processes.

The Birch reduction, discovered more than 80 years ago, has become the standard chemical method for reducing arenes to value-added 1,4-cyclodiene building blocks by alternating electron and proton transfer steps[1,2]. To achieve such challenging dihydro additions to aromatic systems, alkali metals are dissolved in liquid ammonia under cryogenic conditions resulting in the formation of highly reactive solvated electrons (Fig. 1a). Due to the unsustainable reaction conditions, ammonia- and/or alkali metal-free Birch reduction variants[3–8], and, more recently, promising photo- and electrochemical processes have been developed, ushering in a new era in Birch reduction applications[9–11]. An alternative to the classical Birch reduction is provided by dearomatizing benzoyl-coenzyme A (BzCoA) reductases (BCRs) from anaerobic bacteria. BCRs

are key enzymes in the degradation of various homocyclic aromatic compounds derived from lignin and other secondary plant metabolites, aromatic amino acids, or fossil oil-derived products[12,13], and thus play an important role in the global carbon cycle and the elimination of environmentally harmful aromatic compounds. BCRs reduce their substrate to cyclohexa-1,5-diene-1-carbonyl-CoA (dienoyl-CoA), representing an *ortho*- rather than the *para*-dihydro addition of classical Birch reductions. The reaction occurs at a standard redox potential outside the biological redox window (E°′ = −622 mV vs the standard hydrogen electrode at pH 7)[14], which requires a coupling of electron transfer from reduced ferredoxin (Fd$_{red}$) to the aromatic ring with an additional exergonic process.

[1]Faculty of Biology – Microbiology, University of Freiburg, 79104 Freiburg, Germany. [2]Biotechnology Department, Centro de Investigaciones Biológicas Margarita Salas-CSIC, 28040 Madrid, Spain. [3]Department of Molecular Membrane Biology, Max Planck Institute of Biophysics, 60438 Frankfurt am Main, Germany. [4]Faculty of Computational Biochemistry, University of Bayreuth, 95447 Bayreuth, Germany. [5]Department of Chemistry, RPTU Kaiserslautern-Landau, 67663 Kaiserslautern, Germany. ✉e-mail: matthias.boll@biologie.uni-freiburg.de

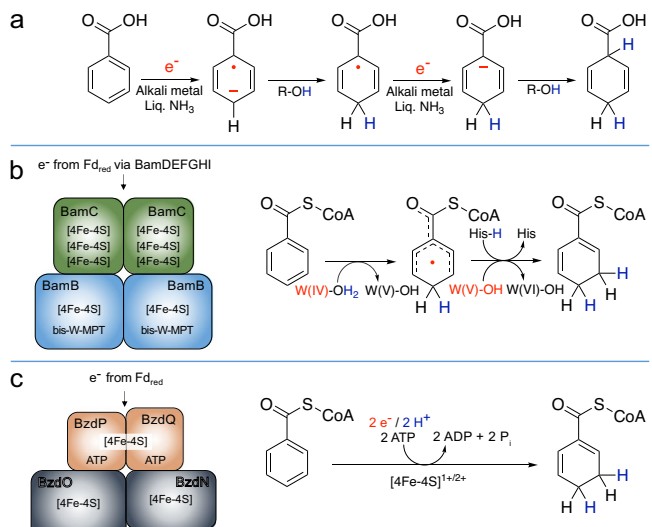

**Fig. 1 | Chemical and enzymatic Birch reduction of benzoic acid/benzoyl-CoA.**
**a** Chemical Birch reduction of benzoic acid through alternate electron and proton transfer steps using alkali metals dissolved in ammonia as electron donor and an alcohol as proton donor. **b** Enzymatic Birch reduction by the active site BamB subunit of class II BCRs through hydrogen atom transfer from an aqua-W-MPT cofactor followed by electron transfer from the same cofactor assisted by protonation involving a conserved histidine. Low potential electrons are provided by the BamCDEFGHI subunits through flavin-based electron bifurcation, with Fd$_{red}$ serving as putative low-potential electron donor. **c** Enzymatic Birch reduction by class I BCRs showing the predicted architecture of the model enzyme from *Aromatoleum* sp. CIB. The electron-activating BzdPQ subunits transfer the energy from ATP hydrolysis to the reduced bridging [4Fe-4S] cluster, which reduces the predicted low-potential [4Fe-4S] clusters of the active site components BzdNO. The BzdPQ and BzdNO modules most possibly form only a transient complex.

Two non-homologous classes of BCRs have evolved in nature that can be distinguished by their subunit composition and energy coupling strategy. Class II BCRs found in strictly anaerobic bacteria are composed of eight subunits that assemble into a one-megadalton enzyme complex Bam[(BC)₂DEFGHI]₂ with more than 50 iron-sulfur (FeS) clusters, 6 flavins and four active site tungsten-*bis*-metallopterin (W-MPT)-cofactors[15,16]. They are proposed to catalyze the endergonic reduction of BzCoA with Fd$_{red}$ as electron donor powered by an energetic coupling to a second exergonic electron transfer from Fd$_{red}$ to an unknown high-potential electron acceptor. This type of energy coupling is achieved at a special flavin cofactor and is referred to as flavin-based electron bifurcation[17]. The catalytic W-MPT-OH₂ cofactor is bound to the BamB subunit and reduces BzCoA via an initial hydrogen transfer from a water ligand yielding a radical intermediate (Fig. 1b). The latter is then reduced to dienoyl-CoA by a single electron transfer from W-MPT, assisted by protonation from a conserved histidine residue[18].

Class I BCRs are commonly found in facultatively anaerobic bacteria and couple BzCoA reduction by Fd$_{red}$ to a stoichiometric hydrolysis of ATP to ADP + P$_i$ per electron transferred. Architecturally, the heterotetrameric enzyme complexes are composed of two functional dimers: the ATP-binding dimer coordinates a subunit-bridging [4Fe-4S] cluster and serves as an ATP-dependent electron activator, whereas the catalytic dimer is predicted to contain an FeS cluster in each subunit and the substrate binding site (Fig. 1c)[19,20]. The best studied representatives of class I BCRs to date are those from β-proteobacterial, denitrifying *Thauera* species[19,20]. The BCR- and MBR (Methyl-BCR)-types of class I BCRs form permanent heterotetrameric complexes, whereas the ATP- and BzCoA-binding dimers from *Aromatoleum species* (formerly *Azoarcus*), referred to as BZD-type (benzoate degradation)[21], are predicted to form only transient complexes[22]. A

BZD-type BCR has not been isolated and studied so far. Class I BCRs belong to the BCR/HAD (2-hydroxyacyl-CoA dehydratase) radical enzyme family[23]. The HADs are involved in the fermentation of amino acids by *Clostridia*, and catalyze the ATP-dependent radical elimination of water from *(R)*-2-hydroxyacyl-CoA to *(E)*-2-enoyl-CoA at an active site [4Fe-4S] cluster[24]. Unlike class I BCRs, HADs catalytically activate a single electron by ATP hydrolysis, which is recycled for multiple turnovers after transfer to the catalytic dimer[25].

In this study, we heterologously produced the active site components BzdNO from *Aromatoleum* sp. CIB and provide a first X-ray structural characterization of the catalytic dimer of a class I BCR with the BzCoA substrate and the dienoyl-CoA product in van der Waals contact with the catalytic FeS cluster. Together with UV-visible and electron paramagnetic resonance (EPR) spectroscopic analyses as well as quantum mechanical/molecular mechanical (QM/MM) calculations, we provide evidence for a Birch-like reduction mechanism of ATP-dependent class I BCRs involving unorthodox FeS clusters for electron transfer and catalysis.

## Results

### Enrichment and general biochemical properties of BzdNO
Previous attempts to structurally characterize tetrameric class I BCRs of the BCR or MBR type have failed. Therefore, we chose the catalytic BzdNO dimer from the β-proteobacterium *Aromatoleum* sp. CIB (WP_050415404.1/WP_224792950.1) as an alternative model to gain structural and mechanistic insights into class I BCR catalysis. It shares only moderate sequence identity with the catalytic subunits of tetrameric BCRs (26%/24% to BCR from *Thauera aromatica* K172, and 24%/25% to MBR from *Thauera chlorobenzoica* 3CB-1)[20]. BzdNO from *Aromatoleum* sp. CIB was anaerobically produced in *E. coli* MC4100 with a C-terminal Strep-tag at the BzdO-subunit and purified by Strep-Tactin® affinity and size exclusion chromatography under anaerobic conditions (Supplementary Fig. 1). The purified protein contained $10.9 \pm 0.6$ mol Fe per BzdNO ($n = 6$; ± standard deviation [SD]), which is significantly higher than the 8 mol Fe expected for the two [4Fe-4S] clusters described for catalytic dimers of BCRs[19,20]. Size exclusion chromatography determined a molecular mass of $107 \pm 3$ kDa ($n = 5$; ± SD) protein suggesting a heterodimeric architecture (theoretical molecular mass ≈ 95 kDa). The purified enzyme complex showed no BzCoA reductase activity with 5 mM Ti-III-citrate or sodium dithionite as an electron donor in buffer at pH 8.9. Instead, it catalyzed the reverse reaction, the methyl viologen-dependent rearomatization of dienoyl-CoA to BzCoA (Supplementary Fig. 2a). Such a reverse reaction was previously described for the catalytic BamBC dimer of the class II BCR[26], but not for the tetrameric class I BCRs from *T. aromatica* or *T. chlorobenzoica*. The highest rate of $5.8 \pm 0.5$ nmol min⁻¹ mg⁻¹ ($n = 3$; ± SD) was observed around pH 8.8 with benzyl viologen as electron acceptor (Supplementary Fig. 2b).

### BzdNO overall X-ray structure reveals similarities to 2-hydroxyacyl-CoA dehydratase and double-cubane cluster binding proteins
The X-ray structure of BzdNO crystals was determined in a BzCoA bound (pdb 8SO2), dienoyl-CoA bound (pdb 8S1T), and in a partially CoA-free state (pdb 8S2R) (for analysis of X-ray data sets, see Supplementary Table 1). Architecturally, the BzdNO complex is composed of the two structurally similar subunits BzdN and BzdO that have a sequence identity of 12.7% and superimpose with a root-mean-square deviation of 2.7 Å (332 of 375 residues) (Fig. 2a).

The closest structurally characterized relatives of BzdNO are 2-hydroxyisocaproyl-CoA dehydratase (HAD, pdb code: 3O3N)[24] and a double-cubane cluster containing protein (DCCP, 6ENO)[27,28] reflected by rms deviations of 2.2 Å (725 of 774) and 3.0 Å (662 of 813) (sequence identity 23.6% and 19.6%), respectively.

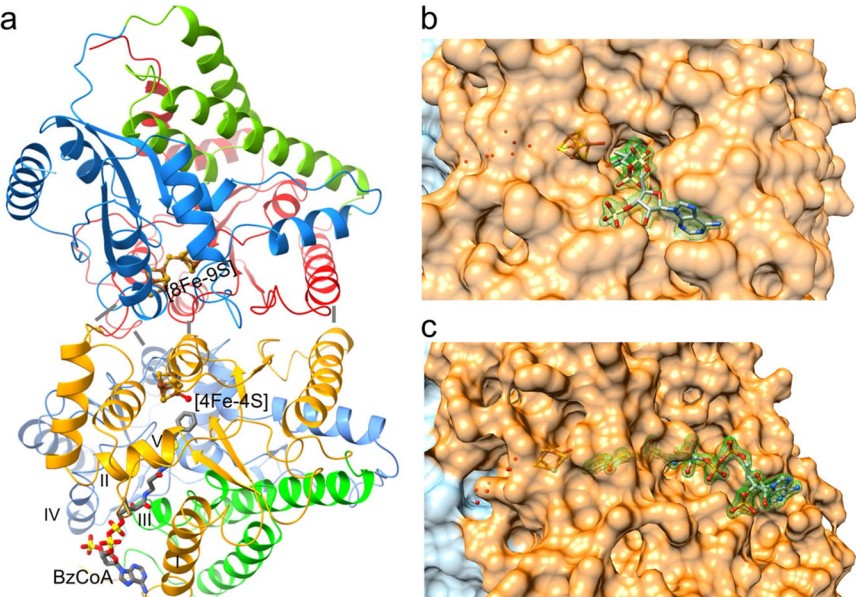

**Fig. 2 | Structure of the BzdNO dimer. a** Structurally similar BzdN (left) and BzdO (right) are composed of an N- (tomato, BzdN; orange, BzdO), and a C- (blue, light-blue) terminal α/β domain as well as an intermediate domain (light-green, green). The FeS clusters (Fe: brown, S: gold) and BzCoA (C: gray) are drawn as ball-and-stick models. Both α/β domains of BzdN and BzdO, especially the tips of several flanking helices of the central β-sheets (gray bars) are the major components of the inter-subunit interface. The N-terminal helices 15:32 (I), 64-70 (II), 220:234 (III) and 281:288 (IV) as well as the loop following strand 269-271 (V) are primarily involved in phosphopantetheine and 3'-phospho-AMP binding. **b** + **c** Global design of the BzCoA binding channel in two different orientations. BzdN and BzdO are drawn as transparent light-blue and orange surfaces, respectively. The channel is about 20 Å long and at its bottom, the BzCoA head is placed in front of the open site of the laterally displaced catalytic [4Fe-4S] cluster. Water molecules, one as a ligand of the [4Fe-4S] cluster, and six forming a chain towards the bulk solvent, are shown with red spheres. The contour level of the $2F_{obs}-F_{calc}$ electron density (green mesh) for BzCoA is 1.6σ.

Both BzdN and BzdO can be divided into three closely attached domains (Fig. 2a). The N- (BzdN: 1–155, BzdO: 1-185) and C- (BzdN: 228–376, BzdO: 264–442) terminal domains each adopt a similar α/β topology with four parallel strands being flanked by two helices on either side. The central strands of both domains are oriented approximately perpendicular and their C-terminal ends contact each other at the top. Three strand-helix linkers of the central α/β-structure essentially form the binding sites for the FeS clusters in BzdN and BzdO, and for the benzoyl thioester head group of BzCoA in BzdO. The intermediate domain (BzdN: 156–227, BzdO: 186–263) consists of three helices that are associated with the N-terminal helix (Fig. 2a). The resulting four-helix bundle occupies the space of a groove formed between the N- and C-terminal domains, thus stabilizing the protein and, in addition, participates in the construction of the active site channel for binding the 4-phosphopantetheine and the 3'-phospho-ADP tail of CoA.

## Binding of two unusual types of FeS clusters

BzdN carries a double-cubane [4Fe-4S]-S-[4Fe-4S], hereafter referred to as [8Fe-9S] cluster (Fig. 2a, Supplementary Fig. 3), which has, so far, only been reported for DCCP. Its specific catalytic function is still unclear[27,28]. The [8Fe-9S] cluster is embedded in a largely encapsulated cavity accessible only via a narrow water-filled channel along Ser239 and Gln324 (Fig. 3a). The two [4Fe-4S] sub-clusters are twisted against each other, and are linked by a sulfur ligand positioned between their two closest irons (3.5 Å Fe-Fe distance). Refinement with a bridging chloride instead of a sulfur resulted in a clearly higher B-factor. The six further irons of the [8Fe-9S] clusters are ligated by cysteine thiolates (Fig. 3a). Significantly different amino acid surroundings characterize the two [4Fe-4S] sub-clusters. The distal [4Fe-4S] sub-cluster (further away from the FeS cluster of BzdO) interacts via its inorganic sulfurs with three arginines (Arg53, Arg267, Arg290), one histidine (His330) and three neutral hydrogen bond-donating side chains (Fig. 3a). This rather positively charged environment implicates a higher redox potential of this [4Fe-4S] sub-cluster. In contrast, the surrounding of the proximal [4Fe-4S] cluster

is more hydrophobic. The [8Fe-9S] cluster of DCCP shows a related polypeptide surrounding except for a reduced number of positively charged residues around the distal [4Fe-4S] sub-cluster.

BzdO carries one [4Fe-4S] cluster that is positioned deeply inside the protein (Figs. 2b, 3b and Supplementary Fig. 3) at a similar place between the α/β domains as described for HadB (2-hydroxy-isocaproyl-CoA dehydratase) and the proximal [4Fe-4S] sub-clusters of the double-cubane clusters in BzdN and DCCP. Interestingly, the [4Fe-4S] cluster of BzdO is accessible to bulk solvent from either a water chain or the substrate binding site; both are occluded in the active site either upon binding of the proposed ATP-activating BzdPQ dimer[28] and the substrate, respectively (Figs. 2b, 3b). The four irons are coordinated to three cysteines (Cys95, Cys128 and Cys380) and a small non-proteinogenic ligand assigned as water ($-OH_2$) or hydroxy ($-OH$) group indistinguishable at the current resolution (Fig. 3b). Sulfur and chloride can be definitively excluded due to the high resolution of the electron density map (Supplementary Fig. 3). The environment of the [4Fe-4S] cluster is rather balanced between polar and non-polar residues; there are two polar inorganic sulfur – polypeptide interactions (Fig. 3b). HadB contains the same [4Fe-4S] cluster coordination as BzdO, however, in a slightly more hydrophobic protein environment (e.g. His131 in BzdO is replaced by an alanine in HadB) and a blocked water channel.

Most interestingly, the $H_2O/OH$ ligand of the [4Fe-4S] cluster pointing to the substrate binding site, is positioned next to the Glu65 carboxylate and the His131 imidazole groups both protruding from the N-terminal α/β domain. Their side chains are fixed and their $pK_a$ values are adjusted via multiple interactions with the polypeptide (Fig. 3b): Glu65-OE1 is hydrogen-bonded with Ala45-N, Glu65-N and a solvent next to Gln124-NE2 and Glu65-OE2 with the $Fe-OH_2/OH$ and Ser44-OG. His131-ND1 interacts with Cys128-N, Ile127-N and weakly with Thr125-O and Cys128-O whereas His131-ND2 with $Fe-OH_2/OH$. Thus, the rather hydrophobic surrounding and the hydrogen-bond pattern are best compatible with uncharged states of the acid/base catalysts Glu65 and His131.

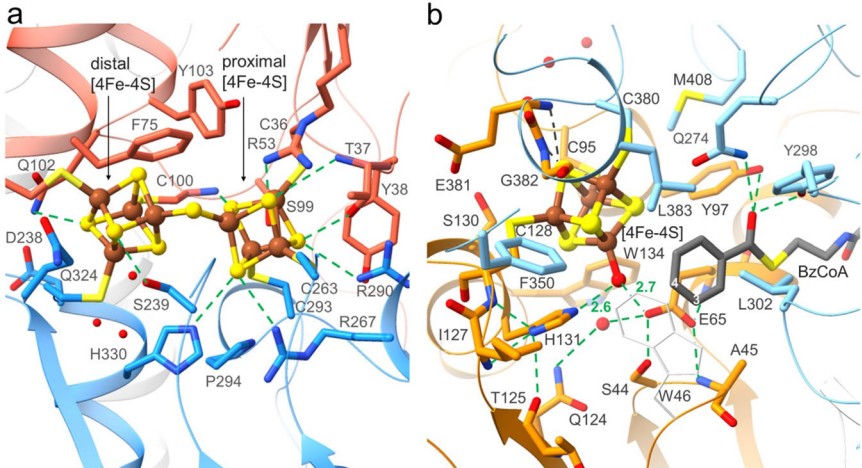

**Fig. 3 | Functionally relevant regions involved in cluster and BzCoA binding.**
**a** Binding site of the [8Fe-9S] cluster in BzdN. The [8Fe-9S] cluster hosts in a cavity between the N (tomato)- and C (blue)-terminal α/β domains formed by loops preceding the first three strands of the N- and C-terminal domain, the N-terminal end of helices 74:85, 103:113, 327-343 and the loop between helices 278:288 and 302:314. **b** Binding site of the catalytic [4Fe-4S] cluster/benzoyl pair in BzdO. The catalytic [4Fe-4S] cluster and benzoyl head (gray) of BzCoA are embedded in a

similar manner between the N- and C-terminal α/β domains (orange and light-blue) in a similar manner as the [8Fe-9S] cluster in BzdN. The planar aromatic ring is sandwiched between Leu302 and the hydrophobic part of Glu65. For an unobstructed view the bonds of Trp46 at the front side are drawn thinner and in white. Hydrogen-bonds are given as green dashes; interactions between Glu381 and Gly382 and the [4Fe-4S] cluster as grey dashes.

## Non-covalent substrate/product binding

BzCoA is embedded inside the active site channel of BzdO in an elongated fashion from the buried benzoyl head to the diphosphate of CoA at the channel entrance (Fig. 2b). After the diphosphate, CoA is kinked by ca. 90° and the 3'-phosphorylated adenosine tail is attached along the protein surface. While the phosphopantetheine moiety is flanked by helices 15:32, 64:70 and 220:234 as well as the loop segment following strand 269:271, the 3'-phosphorylated adenosine moiety is contacted by the N-terminal arm and helices 15:32, 220:234 and 281:288 (Fig. 2a). The detailed substrate-polypeptide interactions are schematically shown in Supplementary Fig. 4.

The benzoyl ring of CoA is laterally displaced relative to the [4Fe-4S] cluster, such that the $OH_2$/OH group points to on one side slightly above the benzoyl plane of the substrate (Fig. 2b). Trp46, Leu302, Leu383 and Tyr97 protruding from the N- and C-terminal α/β domains create a predominantly hydrophobic environment of the benzoyl ring (Fig. 3b). Access of bulk solvent to the active site is completely blocked. While C1, C2 and C6 of the benzoyl ring are not directly adjacent to protonable residues, C4 is 3.1 Å apart from the [4Fe-4S]-$OH_2$/OH group and C3 is 3.6–4.0 Å from the [4Fe-4S]-$OH_2$/OH group, Glu65-OE2 and Ser44-OG. The thioester oxygen of BzCoA is hydrogen-bonded with Tyr97, Tyr298 and Gln274 suitable for stabilizing negative charges by hydrogen-bond interactions (Fig. 3b).

The structure of BzdNO in complex with the product dienoyl-CoA reveals the identical binding mode as the substrate in the BzdNO-BzCoA complex. The dearomatized ring of dienoyl-CoA remains nearly planar; only the C4 sits slightly outside the plane (Supplementary Fig. 5). A structural characterization of BzdNO without previous incubation with substrate/product does not reveal a completely empty substrate binding site but appears to contain besides several solvent molecules a weakly occupied CoA ligand. It is worth to note that in this structure, the [4Fe-4S] cluster binds its $H_2O$/OH ligand at complete occupation, which rules out that Glu65 replaces this ligand in the absence of the substrate/product. Comparison between the enzyme in a CoA ester-bound and partly CoA-free state indicates an induced-fit process upon substrate binding, primarily implicating an inwards shift of helix 281:288 and the preceding loop up to maximally 5 Å. This leads to a narrower active site channel and a rigidification of contacting segments as a result of optimized polypeptide-CoA interactions (Supplementary Fig. 6).

## FeS cofactors exhibit unusual UV-visible and EPR spectroscopic features

BzdNO in buffer (20 μM) exhibited an ultraviolet-visible (UV-vis) absorbance spectrum in the 350–900 nm region that was bleached upon stepwise addition of 5–100 μM sodium dithionite or dienoyl-CoA (pH 9, Supplementary Fig. 7). The oxidized minus reduced difference spectra in the 400–500 nm region revealed a typical peak of [4Fe-4S] clusters. The maximum shifted from 424 to 412 nm (dithionite reduction) and from 427 to 407 nm (dienoyl-CoA reduction) during the titration, indicating the successive reduction of two redox species (Fig. 4a, b). Unusual for [4Fe-4S] cluster difference spectra, a broad shoulder emerged in the 600–900 nm region. Its maximum at 660 nm was more pronounced during the titration with dienoyl-CoA than with dithionite. During the titration with dienoyl-CoA, an almost linear correlation of electron addition/increase of difference spectra absorption was observed up to the addition of three electron equivalents per protein ( ≈ 75% of the maximal reduction) (Fig. 4d). Complete reduction (denoted as 100%) was only achieved with high excess of dienoyl-CoA. With dithionite, the same maximal degree of reduction was achieved albeit at much higher concentrations, and the reduction followed a linear slope only until the addition of the first electron equivalent per protein (Fig. 4c). The different titration behavior of dienoyl-CoA and dithionite can be explained by their different redox potentials of <−700 mV for the BzCoA/dienoyl-CoA[14] and ≈ −550 mV for the sulfite/dithionite[29] pair at pH 9. In summary, the [4Fe-4S]-$OH_2$ and double-cubane of BzdNO appear to be reduced by three electrons. The first electron transfer occurs at a potential > −450 mV, while the second and third require very low redox potentials below −600 mV. At high excess of reductant, even a fourth electron appears to be accepted by the redox cofactors of BzdNO. The nature of the broad shoulder between 600–900 nm, which is not observed for canonical [4Fe-4S] clusters, is attributed to the unusual electronic properties of the double-cubane cluster. It is noteworthy that in the thionine-oxidized state of nitrogenase P clusters, strong MCD bands between 600 and 1000 nm have been described, indicating absorption in this region[30]. During the titrations of the DCCP double-cubane clusters, only a weak shoulder was observed[27]. In the study with DCCP, the titration was carried out at pH 8, thus the necessary low redox potential may not have been reached.

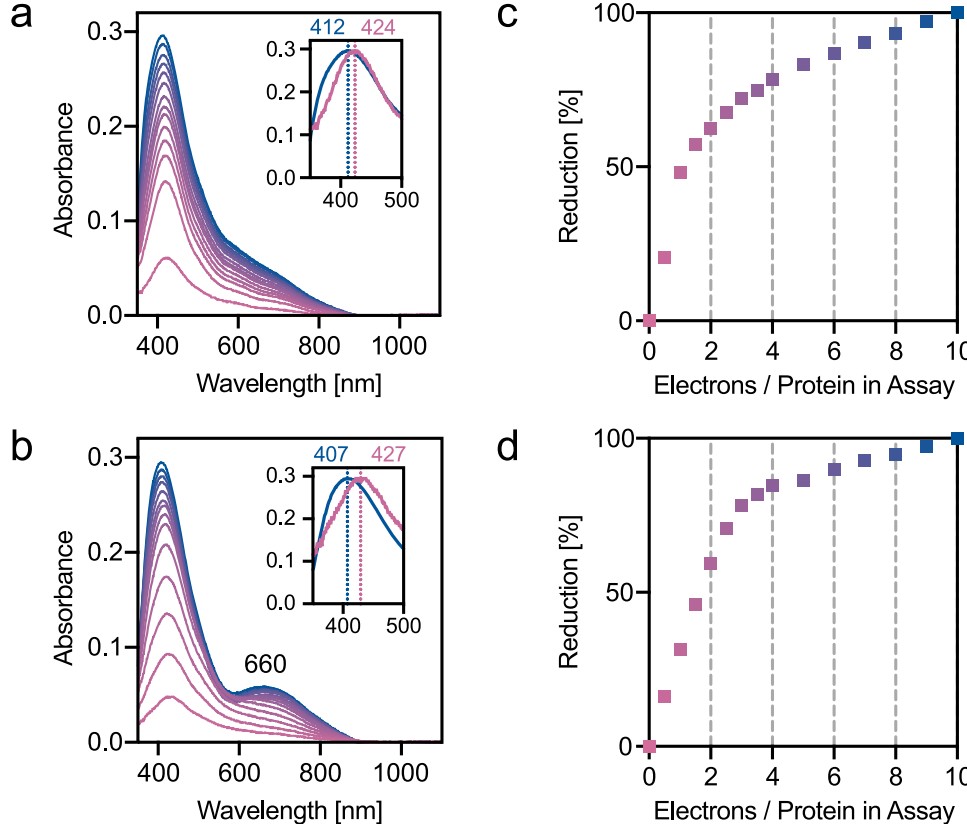

**Fig. 4 | UV-vis spectra of BzdNO.** UV-vis difference spectra of BzdNO (20 µM) reduced with increasing amounts of **a** sodium dithionite or **b** dienoyl-CoA (5 µM steps: 5–40 µM; 10 µM steps: 40–100 µM each) at pH 9 (oxidized minus the stepwise reduced spectra are shown). The original spectra are presented in Supplementary Fig. 7. The insets show a normalized magnification of the 400 nm region at

reducing agent concentrations of 5 µM (magenta) vs 100 µM (blue); the shift of the maxima from 424/427 nm to 412/407 nm is indicated by dotted lines. The degree of reduction is plotted against the number of electrons added during the titration with **c** dithionite and **d** dienoyl-CoA. Source data are provided with this paper.

The FeS clusters of 30 µM BzdNO were further investigated by X-band electron paramagnetic resonance (EPR) spectroscopy. The use of such low enzyme concentrations allowed for the parallel analysis of samples by both EPR (Fig. 5a, b) and UV-vis absorbance spectroscopy (Supplementary Fig. 8). EPR spectra of thionine-oxidized BzdNO at pH 7.8 and 9 showed only a commonly observed signal around $g = 4.3$ for adventitiously bound $Fe^{3+}$ impurities, but no paramagnetic species as expected for a superoxidized $[4Fe\text{-}4S]^{3+}$ cluster. Hence, both the $[4Fe\text{-}4S]\text{-}OH_2$ and the double cubane cluster appeared to be in a fully EPR silent state. BzdNO samples (30 µM) were reduced with both dithionite and dienoyl-CoA (300 µM each), and measured at 10 K in the 600-1800 Gauss and at 20 K in the 3000-4000 Gauss region. At pH 7.8, dithionite reduction resulted only in slight bleaching of the UV-vis absorbance spectrum in the typical [4Fe-4S] cluster region (Supplementary Fig. 8c). The 20 K EPR spectrum of this sample showed a signal typical for an $S = 1/2$ species of a $[4Fe\text{-}4S]^{1+}$ cluster with $g$-values around 2.03/2.01 and 1.90, while no additional signal was observed in the high-field region. At pH 9, an almost threefold bleaching of the UV-vis absorbance spectrum was observed upon dithionite addition, accompanied with the appearance of a weak $g = 5.19$ signal and stronger features in the low-field region at $g$-values between 4.8 and 5.2, typical for an $S = 3/2$ species with high rhombicity (Supplementary Fig. 8d and Fig. 5a, b). In parallel, the $S = 1/2$ signal changed its shape, probably as a result of a magnetic coupling with a second paramagnetic species. We attribute the dithionite-induced signal at pH 7.8 in the $g = 2$ region to the reduced distal sub-cluster of the double-cubane, since its reduction appears to be thermodynamically more feasible than the other clusters due to its rather positively charged environment. The rise of the weak signals at pH 9 may be assigned to the partial reduction of either the

second double cubane sub-cluster or the $[4Fe\text{-}4S]\text{-}OH_2$ cluster. In the presence of dienoyl-CoA at pH 7.8, the bleaching of the UV-vis absorbance spectrum was clearly more pronounced than with dithionite (Supplementary Fig. 8e, f). In parallel, an EPR signal with sharp peaks at $g = 5.68$ and 5.28 emerged that clearly differed from that observed with dithionite at pH 9.0. The two peaks are assigned to an $S = 3/2$ system with a rhombicity of ~0.26 of which only the highest $g$-value of the two doublets is detectable due to the anisotropic nature of the $g$ tensor. In the $g = 2$ region, a highly complex signal appeared, that shared the characteristic features of the $S = 1/2$ signal observed with dithionite at pH 9. Since dienoyl-CoA transfers electrons to BzdNO via the active site cluster, we assign the features at $g = 5.69$ and 5.28 to the reduced $[4Fe\text{-}4S]\text{-}OH_2$ cluster in the benzoyl-CoA-bound state as they were not observed with dithionite at pH 9. Consistent with the observed linear reduction of BzdNO by dienoyl-CoA up to three electron equivalents (Fig. 4d), the active site cluster reduction was most likely accompanied by the reduction of the double-cubane with two electrons, most likely reflected by the unusual absorption shoulder at 660 nm (Fig. 4b). The complex features in the $g = 1/2$ region may result from interactions between the [8Fe-9S] and active site clusters. Surprisingly, both the EPR signal in the $g = 5$ and $g = 2$ region decreased in the dienoyl-CoA-reduced sample at pH 9 when compared to the sample taken at pH 7.8, although the UV-vis spectrum showed a minor further bleaching. This result suggests that reduction by dienoyl-CoA at pH 9 resulted in a partially EPR silent super-reduced state. This interpretation would be consistent with UV-vis absorption spectroscopy data in the presence of excess dienoyl-CoA, which indicated the non-stoichiometric reduction of the $[4Fe\text{-}4S]\text{-}OH_2$ and double cubane clusters by a fourth electron. The highly complex EPR signals obtained with dienoyl-CoA cannot be

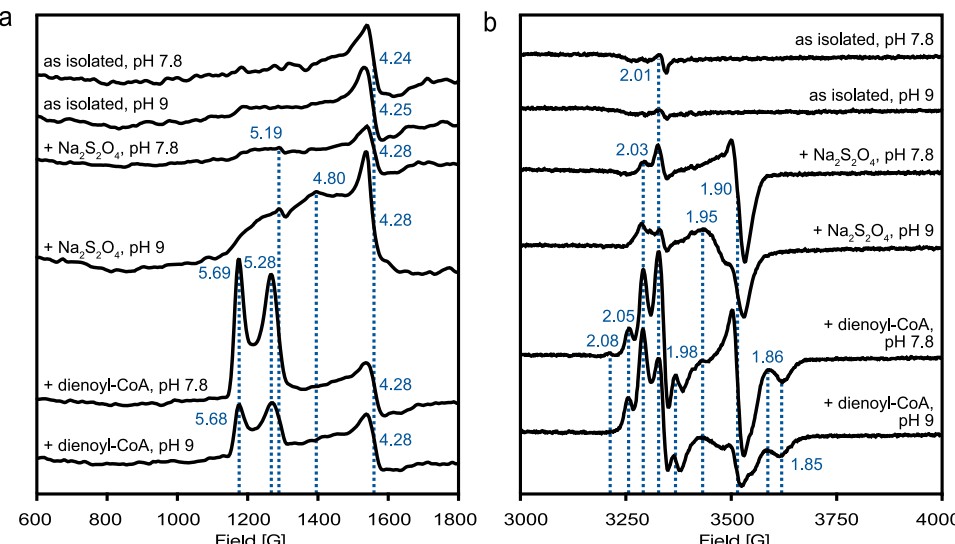

**Fig. 5 | EPR spectra of BzdNO.** The X-band EPR spectra of 30 μM BzdNO are shown in the **a** low-field and **b** high-field region at pH values as indicated: as isolated, in the presence of 300 μM sodium dithionite, and reduced with 300 μM dienoyl-CoA. Selected g-values are given and marked by dotted blue lines. EPR conditions: temperature, 10 K (**a**)/20 K (**b**); microwave power: 21 mW (**a**)/0.008 mW (**b**); modulation amplitude 1.5 mT (**a**)/0.8 mT (**b**); modulation frequency 100 kHz; microwave frequency 9.36 GHz. Source data are provided with this paper.

unambiguously assigned to individual clusters at this stage. To achieve such an assignment, molecular BzdNO variants containing only one of the two FeS clusters of BzdNO would be required. To date, the formation of such variants, based on the replacement of cysteines coordinating the [8Fe-9S] cluster (Cys36/73/100/263/293/327) or the [4Fe-4S] cluster (Cys95/128/380) by alanine, has not yet been successful due to their instability.

### Structure-based QM/MM calculations are consistent with a radical mechanism involving a hydrogen atom transfer step

The structure of BzdNO with the bound BzCoA substrate was used as a starting model for quantum mechanics/molecular mechanics (QM/MM) calculations. The benzoyl-thioester moiety of BzCoA, the [4Fe-4S]-cluster with its water ligand, the side chains of Cys95, Cys128, Cys380, His131, Glu65 and Trp46 constitute the QM region. Most side chains were truncated between Cα and Cβ, only the one of Glu65 was truncated between Cβ and Cδ. Based on structural data, Glu65 was set protonated, whereas His131 was set to its neutral form (protonated at NE2). Starting from a reduced [4Fe-4S]$^{1+}$ cluster, a hydrogen atom of the [4Fe-4S] cluster-coordinating water molecule was moved to C4 of the benzene ring (Fig. 6 upper panel). The activation energy of 6.8 kcal mol$^{-1}$ is sufficiently low to consider the coupled electron/proton (hydrogen atom) transfer process as plausible. This energetically demanding first hydrogen atom transfer step generates a semiquinone radical intermediate that is stabilized by delocalization over the six-membered ring and the thioester oxygen; hydrogen bonds between the latter and Tyr97, Tyr298 and Gln274 (Fig. 3b) further reduce the transition state energy.

In the next step, two alternative scenarios were investigated. First, the second electron transfer starts from the reduced [4Fe-4S]$^{1+}$ cluster after previous reduction from the reduced double-cubane cluster. Transfer of a second electron results in the reduction of the neutral radical to the dienoyl-CoA anion intermediate. In an alternative scenario, the second electron originates from the oxidized [4Fe-4S]$^{2+}$ intermediate which would then be oxidized to the 3+ state. Judging from the energy profiles of the QM/MM calculations, both reaction scenarios with activation energies of 2.3 and 6.6 kcal mol$^{-1}$ are possible and may even occur in parallel. For the reverse reaction from dienoyl-CoA to BzCoA, however, only the first mechanism is plausible since a [4Fe-4S]$^{3+}$ cluster is rapidly reduced by the double-cubane cluster.

Furthermore, oxidation of BzdNO with thionine showed no EPR signals of a paramagnetic [4Fe-4S]$^{3+}$ cluster.

For both scenarios, a transition state search was performed to move the proton from Glu65 to C3 of the dienoyl-CoA anion intermediate. QM/MM calculations indicate a conformational change of the Glu65 side chain after the initial hydrogen transfer, which facilitates protonation to C3 (Supplementary Movie 1/Supplementary Fig. 9). Taken together, the mechanisms for achieving benzoyl-CoA reduction to a dienoyl-CoA by class I and class II BCRs are very similar and proceed via hydrogen atom transfer from metal-OH$_2$ moieties to the aromatic ring yielding a neutral radical intermediate (Fig. 6).

## Discussion

The bacterial degradation of structurally diverse aromatic compounds in anaerobic environments converges to the central metabolite benzoyl-CoA. For the subsequent Birch reduction-like dearomatization step, two distinct enzyme machineries have evolved according to the different energy yields and metal availability. The strategies developed to lower the redox potential from the external donor Fd$_{red}$ for electron transfer to an aromatic ring system and to stabilize radical reaction intermediates are fundamentally different.

Class I BCRs are found in facultatively anaerobic bacteria with an energy metabolism efficient enough to couple BzCoA reduction to a stoichiometric ATP hydrolysis. The hydrolysis of one ATP per electron transferred theoretically allows for a decrease of the redox potential of about 500 mV (with $\Delta G = -n\,F\,\Delta E$, and $\Delta G'$ of ATP hydrolysis ≈ −50 kJ mol$^{-1}$). Thus, with Fd$_{red}$ as donor ($\Delta E' \approx -500$ mV), redox potentials as low as −1 V may be reached. In contrast, strict anaerobes using class II BCRs have much lower ATP yields and drive aromatic ring reduction most likely by a flavin-based electron bifurcation mechanism that lowers the redox potential of the ultimate electron donor (previously reduced by external Fd$_{red}$) maximally to about −0.7 V[18]. As a consequence of the higher potential of the ultimate intrinsic electron donor in class II BCRs, electron transfer to the aromatic ring requires a much more efficient stabilization of low-potential reaction intermediates than in class I BCRs. This is achieved by the use of a complex W-MPT cofactor with its higher delocalization capabilities in class II BCRs compared to the [4Fe-4S]-OH$_2$ cluster in class I BCRs[18]. Accordingly, a radical species was detected in the steady state of class II BCR catalysis, whereas no such species was identified in class I BCRs in the present work.

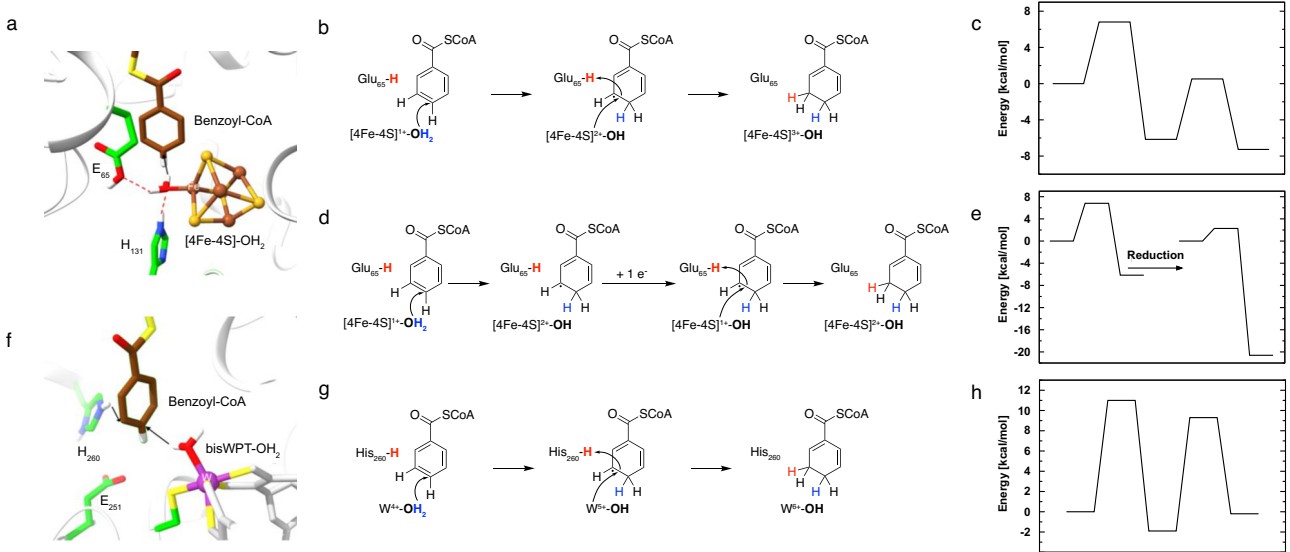

**Fig. 6 | Mechanism of enzymatic Birch reduction by class I and class II BCRs based on X-ray structural data and QM/MM calculations. a** Structure of the active site of BzdO (class I BCR). **b–e** QM/MM calculations-based mechanism with the corresponding energy profiles (for details see Supplementary Table 2-4). In **b** and **c** the oxidized [4Fe-4S]$^{2+}$ cluster serves as electron donor for reduction of the neutral radical prior to re-reduction by electrons from the double cubane cluster; in **d** and **e**, the [4Fe-4S] cluster is reduced prior to the second electron transfer to the neutral radical. **f** Structure of the active site of BamB (class II BCR)[16]. **g**, **h** QM/MM calculations-based mechanism with the corresponding energy profile[18,32] (MM-method: Force Field Charmm36; QM-method: TPSS functional, def2-TZVP basis set).

Despite the differences in electron activation and radical intermediate stabilization, class I and class II BCRs show striking similarities in terms of the catalytic mechanism of aromatic ring reduction at a metal-OH$_2$ device[18,31]. Upon BzCoA binding, members of both BCR classes initiate aromatic ring reduction by hydrogen atom transfer from the reduced metal-OH$_2$ cofactor to the C4 of the benzoyl ring system, followed by a second electron transfer from the assumed reduced metal-OH species. QM/MM calculations clearly showed that the transfer of the hydrogen atom yielding a neutral radical is the rate-determining reaction step of both class I (this work) and class II BCR catalysis[18]. The second electron transfer yielding a dienoyl anion and its subsequent protonation is energetically favored. A neutral radical intermediate is also formed in the classical chemical Birch reduction[2], but rather in two steps via an initial one-electron transfer yielding a radical anion intermediate followed by its subsequent protonation. Such a one-electron reduction without assisted protonation is unlikely to occur in an enzyme, because the redox potential is about −1.9 V (gas phase vs. standard hydrogen electrode)[32], which is far outside the redox window of biological systems. Whereas in class II BCRs, a conserved histidine serves as donor for the exergonic dienoyl-CoA anion protonation[16], in the class I BCR system this step is most likely achieved by a conserved glutamate after a side chain rearrangement upon the first hydrogen atom transfer. Due to the lack of suitable hydrogen atom and proton donors contacting ring carbon positions other than C3 and C4, further reduction of the product is excluded. This selectivity is crucial, because only the dienoyl-CoA formed, but neither a monoenoyl- nor hexanoyl-CoA is an intermediate of the benzoyl-CoA degradation pathway in denitrifying bacteria[33]. Taken together, class I and class II BCRs share the basal chemical conception for reductive dearomatization suggesting an event of convergent evolution as e.g., postulated for the cofactors of H$_2$ activation in the three hydrogenase types[34].

Structural and functional analyses of BCR I/HAD family members (BCR, HAD, DCCP and in part 4-thiouridine-5′-monophosphate desulfidase) provide insights into a divergent evolution process (Fig. 7, Supplementary Fig. 10; for the underlying sequence comparisons see Supplementary Figs. 11 [BzdN] and 12 [BzdO]). Accordingly, the homodimeric DCCPs carrying a double-cubane cluster per monomer represent the archetypes of the BCR I/HAD family[27]. They catalyze the reduction of some small molecules such as acetylene, hydrazine or azide but not N$_2$, although the physiological relevance of these reactions is still unclear[27]. Heterodimeric BzdNO, representing a first example of a double-cubane containing enzyme with a defined catabolic function, may have evolved from a DCCP-like precursor by gene duplication. However, the double-cubane in BzdN converted from an active site to an electrons transfer/storage cofactor. EPR- and UV-vis spectroscopic analyses indicate that it can be reduced by at least two electrons, and probably super-reduced by a third one. It is therefore plausible that the [8Fe-9S] cluster accumulates at least two electrons from the ATP-dependent electron activation module. The catalytic [4Fe-4S]-OH$_2$ cluster most likely evolved from the loss of a [4Fe-4S] sub-cluster from a primordial double-cubane cluster. The space of the lost [4Fe-4S] sub-cluster is occupied by Trp46 and the benzoyl ring of BzCoA. In heterodimeric HADs, both [8Fe-9S] clusters of DCCPs are abolished, and like BzdO, HAD contains a catalytic [4Fe-4S]-OH$_2$ cluster[24]. Although HADs formally catalyze a non-redox reaction, a mechanism involving radical intermediates such as ketyl radicals has been proposed[25]. The second BzdN-like subunit of the HAD dimer hosts a [4Fe-5S] cluster, which can be interpreted as a remnant of the [8Fe-9S] cluster still containing the bridging sulfur atom[24]. Finally, the 4-thiouridine-5′-monophosphate desulfidase TudS consists of only one domain of the BCRI/HAD family subunit and is therefore only moderately similar to other family members. Its only cofactor is a [4Fe-5S] cluster as present in HAD[35,36], and it appears to be functionally transformed from a radical-generating to a sulfur-transferring enzyme. Taken together, these examples demonstrate the versatility of the BCR I /HAD family based on the elimination of a FeS sub-cluster from the double-cubane containing precursor thereby creating novel catalytic functions.

The identification of distinct active site metal-OH$_2$ centers in unrelated class I and II 'Birch reductases' provides a promising starting point for the synthesis and application of biomimetic systems for alternative Birch reduction processes. The current limitation of enzymatic Birch reduction is the restriction to CoA ester substrates.

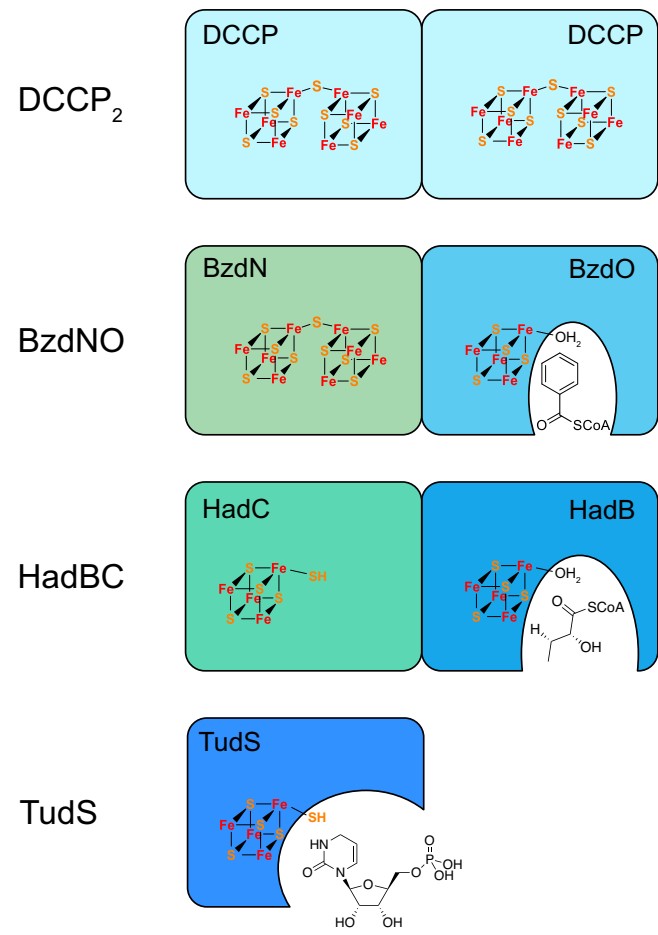

**Fig. 7 | Proposal for the gain of catalytic function as a result of the loss of a sub-cluster of a [8Fe-9S] cluster.** Two [8Fe-9S] clusters in each of the homologous subunits have been identified only in archetypical DCCPs. The biological function of the latter is unknown.

Although challenging, the development of alternative, bioinorganic synthetic CoA-independent substrate activation strategies, and their coupling with already available ammonia-free electron donor systems will open new perspectives for future applications.

## Methods

### Bacterial strains and plasmids

*Escherichia coli* MC4100 was used for heterologous production of BzdNO. The strain was transformed with the plasmid *pIZ_bzdNO(St)* encoding BzdNO (RefSeq WP_050415404.1/WP_224792950.1) with a C-terminal Strep tag at BzdO under the control of the Ptrc-promotor. Plasmid variants encoding cysteine-to-alanine replacement mutants of BzdNO were generated via fast cloning[37] and confirmed by sanger sequencing (Eurofins Genomics).

### Heterologous production of BzdNO in *Escherichia coli* MC4100

*E. c.* MC4100 /*pIZ_bzdNO(St)* was cultivated in buffered TB medium at pH 7.5 (50 mM fumaric acid, 12 g L$^{-1}$ tryptone, 24 g L$^{-1}$ yeast extract, 2.5 g L$^{-1}$ glycerol, 0.76 mM Fe-(III)-citrate, 15 mM monopotassium phosphate, 74 mM dipotassium phosphate, 6.5 g/L potassium hydroxide), supplemented with gentamycin (0.02 mg mL$^{-1}$) in 2 L bottles. Following inoculation, the headspace was flushed with nitrogen gas, the bottles were sealed with gas-tight rubber stoppers and incubated at 37 °C and 80 rpm. Protein production was induced in the exponential growth phase by Isopropyl-β-D-1-thiogalactopyranosid (0.75 mM) and changing the incubation temperature to 25 °C. In addition, cells were supplemented upon induction with 3 mM sodium nitrate, 0.8 mM magnesium sulfate, 0.1 mM calcium chloride, trace elements, and a vitamin solution. Approximately 20 h after induction, cells were harvested anaerobically by centrifugation at 6000 × *g* / 4 °C and stored in sealed anaerobic bottles at −80 °C.

### Enrichment, molecular mass and iron content of BzdNO

Protein enrichment was performed under anaerobic conditions at 4–10 °C. *E. c.* MC4100 / *pIZ_bzdNO(St)* cells were resuspended in 20 mM MOPS/KOH (3-(N-Morpholino)-propanesulfonic acid) at pH 7.8, supplemented with 1 mM dithioerythritol, 0.05 mM sodium dithionite and a pinch of lysozyme, DNase I, and RNAse A. The cell suspension was passed through a French pressure cell at 7.6 MPa, followed by ultra-centrifugation at 150.000 *g* for 1 h. The supernatant was filtered (0.45 μm) and BzdNO was separated from cell extract via Strep Tactin® XT 4Flow® protein purification on an Äkta pure™ FPLC system (Cytiva), equilibrated with 20 mM MOPS/KOH pH 7.8 supplemented with 1 mM dithioerythritol and 0.05 mM sodium dithionite. The bound protein was washed with multiple column volumes of the same buffer and the protein was eluted from the resin with 20 mM MOPS/KOH pH 7.5 supplemented with 50 mM D-biotin, 1 mM dithioerythritol, and 0.05 mM sodium dithionite. The enriched fractions were concentrated with Vivaspin™ Turbo 15 RC 50000 MWCO concentrators (Sartorius) and desalted on a HiLoad® 26/600 Superdex® 200 pg gel filtration resin (Cytiva) equilibrated with 20 mM MOPS/KOH pH 7.5 supplemented with 5 % (w/v) glycerol, 1 mM dithioerythritol, and 0.05 mM sodium dithionite. The gel filtration fractions were concentrated again and filtered through PD-10 resins (Cytiva), previously equilibrated with 20 mM MOPS/KOH pH 7.5 supplemented with 20 % (w/v) glycerol, 1 mM dithioerythritol, and 0.05 mM sodium dithionite. Purified BzdNO samples were stored in sealed anaerobic bottles at −80 °C. Purity of BzdNO was analyzed by sodium dodecyl-sulfate polyacrylamide gel electrophoresis, and the protein bands of BzdNO were identified from SDS gels by mass spectrometric analyses of tryptic peptides (sequence coverage of 72%/69%; O-/N-subunit). For identification, protein bands were excised from the gel, treated with dithiothreitol and iodoaceta-mide (to reduce and alkylate cysteine residues) and hydrolysed with trypsin (Sigma-Aldrich). Peptide fragments were separated on a Waters Acquity I-class UPLC with a Waters Peptide CSH C18 column (2.1 mm × 150 mm, 1.7 μm particle size) coupled to a Waters Synapt G2-Si HDMS ESI/Q-TOF system using a gradient from 1 to 40% ACN/0.1% formic acid (v/v) in water/0.1% formic acid (v/v) at a flow rate of 0.04 ml$^{-1}$ min$^{-1}$. Detection of protein fragments was performed in positive HD-MS$^{E}$mode, (3 kV cone voltage, 120 °C source temperature, 400 °C desolvation temperature, 800 L h$^{-1}$ N$_2$ desolvation gas flow, 6.0 bar nebuliser pressure). Obtained data was analysed with ProteinLynx Global Server (Waters) by matching with the amino acid sequences of *Aromatoleum* sp. CIB (min. fragment ion matches per peptide = 3, min. fragment ion matches per protein = 7, min. peptide matches per protein = 1, max. false discovery rate 4%), as described[38]. The molar mass of BzdNO was determined by analytical gel filtration on a Superdex 200 Increase 10/300 GL resin (Cytiva) under anaerobic conditions. The resin was previously equilibrated with 20 mM MOPS/KOH pH 7.5 supplemented with 1 mM dithioerythritol and 0.05 mM sodium dithionite. Thyroglobulin (669 kDa), apoferritin (443 kDa), alcohol dehydrogenase (150 kDa), Bovine serum albumin (67 kDa) and carbonic anhydrase (29 kDa) were used as standards for column calibration. BzdNO was analyzed five times to determine the standard deviation. The iron content of the samples was determined by the Ferene-method[39].

### Determination of enzyme activities

All enzyme assays with BzdNO were carried out under anaerobic conditions in a glove box (95% N$_2$, 5% H$_2$). For testing the forward reaction, BzdNO (1.4 mg mL$^{-1}$) was incubated with BzCoA (0.5 mM)

with Ti-III-citrate or sodium dithionite (5 mM each) in the presence or absence of methyl viologen (0.25 mM) in 100 μL N-Tris(hydroxymethyl)methyl-3-aminopropanesulfonic acid (TAPS)/NaOH pH 8.9 (100 mM) at 30 °C. For the reverse reaction, recombinant BzdNO (2 mg mL$^{-1}$), the electron acceptor methyl viologen (0.5 mM) and dienoyl-CoA (0.2 mM) were incubated at 30 °C for 30 min in 500 μL of 100 mM TAPS/KOH pH 8.8. The reactions were stopped by addition of double the volume of methanol. After centrifugations, supernatants were analyzed by an Acquity H-Class ultra performance liquid chromatography device (Waters) using an UPLC BEH C18, 130 Å, 2.1×100 mm column (1.7 μm particle size, Waters); For separation, a gradient of acetonitrile/potassium phosphate at pH 7 was applied. BzCoA and dienoyl-CoA were distinguished by their retention times and spectra compared to standards.

## UV-visible spectroscopy
UV-visible spectra of BzdNO were recorded under anaerobic conditions using a Shimadzu UV-1800 spectrophotometer in a quartz cuvette. Continuous benzyl-viologen reduction assays were performed in quartz cuvettes containing 0.5 mM benzyl-viologen, 0.2 mM dienoyl-CoA and 20 μM BzdNO in five different buffers (100 mM MOPS/KOH pH 7.5 or 7.8; 100 mM TAPS/NaOH pH 8.8; 100 mM N-cyclohexyl-3-aminopropanesulfonic acid (CAPS)/KOH pH 9.8 or 10.3) at 30 °C. Electron transfer from dienoyl-CoA to benzyl-viologen was monitored at 600 nm.

## Crystallization and structure determination
The enzyme solution prepared for crystallization was composed of 20–25 mg/ml BzdNO, 20 mM MOPS pH 7.5, 1 mM DTE, 50 μM sodium dithionite and 20% (w/v) glycerol. For co-crystallization attempts 5 mM BzCoA and dienoyl-CoA were added to the protein solution. Crystallization attempts were performed at 18 °C in an anaerobic tent (95% N$_2$, 5% H$_2$) using an OryxNano robot (Douglas Instruments) for sitting drop vapor diffusion experiments. Best crystals grew by mixing 0.3 μl enzyme solution and 0.3 μl reservoir solution composed of 19% (w/v) PEG 3350, 100 mM BTP pH 7.5 and 200 μM sodium formate or sodium acetate. For vitrification the crystals were quickly incubated in solution in which 19% (w/v) PEG 3350 is replaced by 10% (w/v) PEG 3350 and 30% PEG 400. Data were collected at the Swiss-Light source beamline PXII and processed with XDS/XSCALE[40]. An Alphafold[41] model of a BzdNO dimer was calculated and phases determined by the molecular replacement method using PHASER[42]. Model rebuilding, substrate and solvent incorporation were performed in COOT[43]. The resulting coordinates were refined within PHENIX[44]. The parameter file of benzoyl-CoA was originally taken from a previous file[16] and then further adapted to the highly resolved electron density. Crystal parameters, data collection and refinement statistics are listed in Supplementary Table 1. Composite omit electron density map sections for the three applied BzdNO structures are shown in Supplementary Fig. 13. Figure 2/3/6 were produced with Chimera or ChimeraX[45]. The C2 crystal form of BzdNO has a translational pseudosymmetry of 0, 0, and 0.5, whose degree is different in the collected data sets. For this work we chose the data with the lowest translational component. The p-values of the three applied data sets were below 0.05, which corresponds to a small translational pseudosymmetry[46]. The atomic coordinates and the structure factors of BzdNO have been deposited in the Protein Data Bank, www.pdb.org with the ID codes 8SO2 (BzCoA bound state), 8S1T (dienoyl-CoA bound state) and 8S2R (partially CoA-free state).

## EPR spectroscopy
EPR spectroscopy was performed on an upgraded Elexsys E580 Bruker X-band spectrometer, equipped with a 4122HQE resonator. The temperature of the samples was maintained at 4.8 K–20 K with a continuous-flow helium cryostat (Oxford Instruments ESR 900). A Stinger closed-cycle cryostat (Cold Edge Technologies) and a F-70 Sumitomo helium compressor were used for cooling. BzdNO was desalted and subsequently diluted in 100 mM MOPS/KOH pH 7.8 or 100 mM TAPS/NaOH at pH 9.0. Samples were either directly transferred into EPR tubes and frozen, or previously reduced with sodium dithionite or dienoyl-CoA. For comparison, a BzdNO-sample was oxidized with thionine and desalted in 100 mM TAPS/NaOH pH 8.9 and frozen in an EPR tube for analysis.

## QM/MM calculations
The structure of BzdNO with the bound BzCoA substrate was used as a starting model for the calculations. Internal water molecules were placed using the program McVol[47]. The structural model was prepared using the software CHARMM[48] and the CHARMM force field[49] analogous to the procedure described before[31]. The double-cubane FeS cluster was assumed to be reduced and its partial charges were calculated using the CHELPG method[50] in ORCA[51]. The force field parameters for BzCoA were taken from analogous parameter sets in the CHARMM force field. The QM/MM calculations were performed using ORCA. We used unrestricted DFT as QM method, namely we used the BP86 functional with the def2-SVP basis set for the search of the path. Single-point calculations of the end states and the transition states were done using the TPSS functional and the def2-TZVP basis set. The MM energies were calculated using the CHARMM36 force field[49]. To model the QM/MM boundary, a link-atom scheme and electrostatic embedding was used. The QM region was surrounded by a fully flexible MM layer of 8 Å. The [4Fe-4S]-cluster was treated by the broken-symmetry approach[52]. All transition states were searched by the climbing-image nudged-elastic band approach[53]. Details of the calculations are given in the Supplementary Methods and Supplementary Tables 2–4.

## Reporting summary
Further information on research design is available in the Nature Portfolio Reporting Summary linked to this article.

# Data availability
The protein structural data generated in this study have been deposited in the RCSB Protein Data Bank, www.pdb.org, under accession code 8SO2 (BzCoA bound state; https://doi.org/10.2210/pdb8SO2/pdb), 8S1T (dienoyl-CoA bound state; https://doi.org/10.2210/pdb8S1T/pdb) and 8S2R (partially CoA-free state; https://doi.org/10.2210/pdb8S2R/pdb). The output files of the quantum chemical calculation including the spin populations and the electronic structure (single point calculations of the reactant, product and transition states of the suggested mechanism in various spin states) are available at zenodo (https://doi.org/10.5281/zenodo.14811052). The atomic coordinates of the calculated states is provided in the Source Data file. The mass spectrometry data have been deposited to the ProteomeXchange Consortium via the PRIDE[54] partner repository with the dataset identifier PXD053203. Additional data that support the findings of this study are available from the corresponding author upon request. Source data are provided with this paper.

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

## Acknowledgements

This research was supported by the German Research council within RTG 1976, grant number 235777276 (to J.F. and M.B.) and SPP 1927: Iron-Sulfur for Life - Project number 311144407 (to G.M.U.). The EPR spectrometer upgrade and closed-cycle cryostat (A.J.P.) was funded by the DFG (248/320-1, project number 444947649) and the government of Rhineland-Palatinate. We thank Hartmut Michel for continuous support and the staff at the PXII beamline at the Swiss-Light source, Villigen, Switzerland for help during data collection. The research stay of U.F.A. at the University of Freiburg was funded by the grant PID2022-142540OB-I00 from the Spanish Ministry of Science, Innovation and Universities. M.B. and J.F. thank Eva Steuber, Laura Offermanns and Johanna Schell for screening of cysteine mutants of BzdNO and Tingyi Zhan for assistance during EPR sample preparation.

## Author contributions

J.F. heterologously produced, enriched and biochemically characterized BzdON, prepared and measured samples for UV-vis and EPR spectroscopy; analyzed spectroscopic data, prepared figures and contributed to the writing of the paper; U.F.-A. cultivated *Aromatoleum* strain CIB, performed molecular biological work; U.D. prepared samples and conducted experiments for protein crystallography; E.D. provided resources for microbiological and molecular biological work; M.U. performed QM/MM calculations, prepared figures contributed to the writing of the paper; A.J.P. provided resources for EPR spectroscopy, measured, analyzed, and evaluated EPR spectroscopy data; U.E. conducted X-ray analyses of protein crystals, analyzed data, prepared structural models and figures, contributed to the writing of the paper; M.B. designed the study, analyzed data, wrote the paper. All authors discussed results and reviewed the manuscript.

## Funding

## Competing interests

The authors declare no competing interests.
