## [Transparent Peer Review file · Nature Communications]

Enzymatic Birch reduction via hydrogen atom transfer at [4Fe-4S]-OH₂ and [8Fe-9S] clusters

Corresponding Author: Professor Matthias Boll

Version 0:

Reviewer comments:

Reviewer #1

(Remarks to the Author)

The manuscript by Fuchs et. al. describes the crystallographic and mechanistic analysis of the catalytic subunits from a class I benzoyl-CoA reductases (BCR) from *Aromatoleum* sp. CIB. BCRs play an important role in the global carbon cycle by reducing recalcitrant aromatic compounds found in plant lignin, amino acids and environmental contaminants derived from fossil fuels. The authors provide convincing structural data on the catalytic core of BCR (BznD/O) at a high resolution bound to the substrate benzoyl-CoA, a partially reduced dienoyl-CoA, and substrate free. In the substrate bound structure, the benzyl ring is positioned in close proximity to a water ligand on the active site 4Fe-4S cluster. Using this structure, the authors perform a series of QM/MM calculations to propose a catalytic mechanism for benzyl-CoA reduction. These calculations predict a strikingly similar mechanism to that of the Class II BCRs that utilize a different tungsten based metallo cofactor. Overall, this paper is very well-written and the figures look great. My only major concern is the reliance on calculations to understand the mechanism. Also, as far as I can tell, figure 4 just shows that the enzyme the authors are working with is active and doesn't provide any insight into the proposed unique reaction catalyzed by this class of BCRs. It would have been nice to see some kinetic EPR data to understand the connection between the UV-vis spectra and the reduction of the cluster. It is also not commented on which features in the EPR spectra belong to which cluster(s) in the BznD/O complex. Overall the paper is very exciting and merits publication because it shows that a class I BCR exists that is amenable to study.

Reviewer #2

(Remarks to the Author)

Birch reduction is an important synthetic method for achieving hydro additions to arenes, which typically uses strong reducing agents, such as alkali metals. The biological Birch reduction utilizes a very mild condition and is accomplished by metalloenzymes. Unlike the tungsten-bis-metalloprotein cofactor used in the class II benzoyl CoA reductase, this manuscript uncovered a new strategy used in the class I benzoyl CoA reductase. A non-canonical double-cubane [8Fe-9S] and active site aqua-[4Fe-4S] cluster were identified through X-ray structure analysis. A radical mechanism via a proton coupled electron transfer from the metal-bound water to the substrate was suggested on the basis of kinetic, spectroscopic and QM/MM calculations. The results are of great and broad interest and should be published in *Nat. Commun.* after addressing the following minor issues:

1. The total size and total charge of the QM region should be given.
2. For the iron-sulfur cluster, there are six possible antiferromagnetically-coupled states. Did the author consider one of them, or all of them? The spin populations on each metal ion should be given for all calculated structures (shown in the supporting information). This could be used to verify if the correction state was obtained.
3. Single-point calculations using larger basis sets, such as def2-TZVP, should be performed.
4. BP86 has been known to underestimate the barrier for metalloenzymes, other functionals could be used for single-point calculations. Also, dispersion correction should be added as single-point.
5. What are the protonation states of all titrable residues, such as His, Glu, Asp, Lys Arg? PropKa could be used as a crude estimation.

Reviewer #3

(Remarks to the Author)

Remarks to the author

In the current report, Fuchs et al. describe the first crystal structure of the catalytic module of class I BCR, where each subunit of the heterodimer hosts different FeS clusters, [8Fe9S] cluster for electron transfer/storage and [4Fe4S]-OH₂ cluster for active site. The authors proposed that the active site water molecule functions in a dual role: hydrogen atom and electron donor in the low redox regime, consistent with the Birch reduction of class II BCRs containing W-MPT and supported by kinetic, QM-MM and spectroscopic (EPR & UV-VIS) studies. This work will provide the basis for further mechanistic studies on the BCR as well as on other radical-dependent dehydratases. The manuscript is well written, thoroughly interpreted and presented, but the structure is not clearly presented (see comments below) with such a beautiful structure.

The reviewer strongly recommends this manuscript for publication in Nature Communications.

However, the following minor comments must be taken into account.

General comments:

In general, use one type of writing style to describe atoms in amino acids in the main text as well as in SI and in figures: for example, Nε2 (preferred) or NE2 of His.

Main text:

Line 2: replace "Fe4S4-OH2 and Fe8S9 clusters" with "[4Fe-4S]-OH2 and [8Fe-9S] clusters" for consistency throughout the text.

Line 18: replace "Fe4S4 cluster" with "[4Fe-4S] cluster" for the consistency throughout the text.

Line 132: Based on the data in Fig. S2b, the highest activity was at pH 8.8 rather than around pH 9.

Line 151-153: "The channel is about 20 Å long and at its bottom" is not clear. Where is the bottom? Based on Figure 2b, the 3'-phosphoadenosine and diphosphate moieties are not inside the channel, but in a pocket on the surface. It would be clearer if the figure were in a stereo view or if the channel lengths of BzCoA and water molecules were shown from the surface of the protein.

Line 154: describe what kind of electron density is being displayed? AND replace "green" with "green mesh".

Line 154: replace "marked by" with "shown with".

Line 176: The authors have not provided any experimental evidence for the presence of a double cubane cluster (DCC) other than metal content, such as an omitted electron density together with an anomalous map if possible. It would be ideal to show the electron density for DCC, for example in Fig. 3a, but also in SI. Since the occupancy of the metal clusters can be well calculated at this resolution, the occupancy of the DCC and the active site clusters should be mentioned.

Line 181: although the anionic chloride ion did not fit the mut-S position, what about the lighter atoms such as oxygen?

Line 191: in Fig. 3a indicate which sub-cluster is proximal or distal in the figure.

Line 195: similar to the comments above for DCC, only the model is shown in the figure. Please include electron density maps, at least an omitted map.

Line 205: what could be the role of the water chain, how does it interact with the protein, and is this chain still present in the structure of partially occupied CoA?

Line 207-210: the authors claim that the ligand at the open coordination site of the [4Fe-4S] cluster could be a water or hydroxy ligand, which actually makes sense for the mechanism proposed later, and -SH and Cl⁻ are ruled out because of the high resolution. However, if you play with occupancy for refinement, sulfur or Cl could fit with low occupancy and b-factor. Please comment on this.

Line 212: draw or indicate "two polar interactions" in Fig. 3b.

Line 220: 1) there is no indication of the backbone nitrogen of Glu65 in the figure; 2) the solvent next to Gln124 and Glu65 is not H-bonded to Fe-OH₂/OH and Ser44, where the solvent is actually H-bonded to Gln124 and Glu65, and Ser44 is H-bonded to Glu65.

Line 222: 1) Thr125 is not shown in the figure; 2) replace "-ND2" with "His131-ND2".

Line 238: Tyr95 is not shown in Fig. 3b and Fig. S3. Instead there is Tyr97. It seems that Tyr97 is correct because Cys95 is there. Please check this.

Line 240: It would be better to draw the carbon numbers of the benzoyl ring in Fig. 3b.

Line 243: replace "Tyr95" with "Tyr97".

Line 248: Indicate C4 in benzoyl ring in Fig. S4, making the authors' claim more visible.

Line 251: the authors claim that the crystal structure of BzdNO, free of substrate/product, shows a weakly occupied CoA ligand. The protein was heterologously produced by *E. coli*, purified and crystallized without substrate/product. The reviewer is interested to know where the CoA comes from, because *E. coli* would not produce large amounts of CoA? Second, the crystallization solution contains PEG molecules. If a molecule is partially occupied along with a translational pseudosymmetry character of the data set and diameter of the CoA channel, a short chain of PEG molecules may also be present in the channel. Please comment on this.

Line 255: large conformational change of residues 279-280 including very bulky residues (F, W, L) are moved into the CoA ester binding site to optimize the binding (Fig. S5). Is there any other changes in the structure due to this conformational change?

Line 361: It appears that Tyr95 does not exist and Tyr97 may be correct. Please check the text.

Line 455: Although the sequence similarity is low, the structure of the BzdO subunit is very similar to BzdN, a subunit of DCCP, and also to the HAD subunit with [4Fe-5S] cluster (beta subunit). It would be helpful to understand what has changed in the space of a sub-cluster during evolution by showing a superposition of these structures in SI.

Line 571: As shown in Table S1, the Rwork/Rfree values after refinements are exceptionally high compared to the resolutions. This could be the presence of translational pseudosymmetry as described by the authors. Therefore, it would be better to explain this relationship in more detail in the Methods and report the degree of translational pseudosymmetry for each data set in Table S1.

Supporting Information

Line 5: replace "Fe4S4-OH2 and Fe8S9 clusters" with "[4Fe-4S]-OH2 and [8Fe-9S] clusters" for the consistency throughout the text.

Line 21: incorporate PDB-ID of each data set in table S1.

Line 66: Black dotted lines for hydrophobic contacts are not shown in the figure. One-letter code for amino acids causes some confusion with atom names in the figure. Please use the three-letter code.

Line 73: 1) Describe what kind of electron density is shown; 2) In a, mark C4 to show the out of the benzyl ring plane.

Line 83: The partially CoA-free state is not colored in orange, but some light green with outlined backbone is there. clarify this.

Line 110: Explanation of green and orange meshes is required.

Version 1:

Reviewer comments:

Reviewer #1

(Remarks to the Author)

Great paper. I agree, the experiments I proposed are beyond the scope of this paper.

Reviewer #2

(Remarks to the Author)

The manuscript has been revised significantly according to all reviewers' comments, and it could be published after addressing one minor concern:

1. The calculated relative energies of different spin states and spin populations on each metal ion should be listed as tables in the supporting information for friendly checking. It is very difficult to check the original files on Zenodo as the files are too large to be downloaded (more than 1GB).

Reviewer #3

(Remarks to the Author)

The authors have addressed all concerns raised with the first version of the manuscript.

I am satisfied with the revised version of the manuscript and recommend this version for publication. I congratulate the authors on this wonderful work.

Reviewer #1 (Remarks to the Author):

My only major concern is the reliance on calculations to understand the mechanism. Also, as far as I can tell, figure 4 just shows that the enzyme the authors are working with is active and doesn't provide any insight into the proposed unique reaction catalyzed by this class of BCRs. It would have been nice to see some kinetic EPR data to understand the connection between the UV-vis spectra and the reduction of the cluster. It is also not commented on which features in the EPR spectra belong to which cluster(s) in the BzdNO complex. Overall the paper is very exciting and merits publication because it shows that a class I BCR exists that is amenable to study.

Response:

-Fig. 4 shows all steady-state UV/vis and EPR spectra that could be recorded, and that give insights into the number of electrons taken up by the redox clusters and the spin-states and other features of the FeS clusters. We agree with the reviewer that an unambiguous assignment of signals to either of the two clusters is not possible at this stage. To solve this problem we extended our efforts to specifically modify/destroy one of the clusters by site-directed mutagenesis. However, none of the mutants, where a Cys was exchanged was stable. We stated this as follows at line 332 ff: "The highly complex EPR signals obtained with dienoyl-CoA cannot be unambiguously assigned to individual clusters at this stage. To achieve such an assignment, molecular BzdNO variants containing only one of the two FeS clusters of BzdNO would be required. To date, the formation of such variants, based on the replacement of cysteines coordinating the [8Fe-9S] cluster (Cys36/73/100/263/293/327) or the [4Fe-4S] cluster (Cys95/128/380) by alanine, has not yet been successful due to their instability."

-We agree that pre-steady-state kinetics by freeze quench EPR spectroscopy would be a method to trap possible (radical) reaction intermediates by freezing highly concentrated enzyme preparation that have been reacted with substrates (below the ms scale). However, this would be a study on its own that needs very specialized equipment that's not available in any of the author's labs (worldwide, there are only a very few labs where such an equipment is available). Moreover, any kind of substrate-based signal would require labelled substrates to proof their assignment. And even if FeS signals would time-dependently appear, their assignment would still be unclear. So, it would be a tremendous effort to perform pre-steady state kinetics monitored by EPR spectroscopy with unclear outcome. The current work focuses on the initial structural insights, that were the basis for QM/MM calculations. In the revised version, the QM/MM calculations were extended according to the statements of reviewer #2, and leave little doubt on the proposed mechanism.

Reviewer #2 (Remarks to the Author)

1. The total size and total charge of the QM region should be given.

-The QM region was so far only described in the Discussion. We give now a detailed description is given in the supporting information including the requested information.

2. For the iron-sulfur cluster, there are six possible antiferromagnetically-coupled states. Did the author consider one of them, or all of them? The spin populations on each metal ion should be given for all calculated structures (shown in the supporting information). This could be used to verify if the correction state was obtained.

-Originally only one state was considered. All NEB calculations were done with one spin state. Single point calculations were now done for all possible spin states using the functional TPSS with the basis set def2-TZVP and a dispersion correction (D3). The orca output files including the spin populations are provided on zenodo.

3. Single-point calculations using larger basis sets, such as def2-TZVP, should be performed.

-See answer to remark 2

4. BP86 has been known to underestimate the barrier for metalloenzymes, other functionals could be used for single-point calculations. Also, dispersion correction should be added as single-point.

-See answer to remark 2, The values in Figure 5 are updated.

5. What are the protonation states of all titratable residues, such as His, Glu, Asp, Lys Arg? PropKa could be used as a crude estimation.

-In the NEB calculations, all titratable residues are set to their default protonation. Following the suggestion of the reviewer, we calculate the protonation probability of the titratable residues. The program PropKa is known to have problems with residues near metal centers. We therefore used a method that is based on Poisson-Boltzmann equation using our own software GMCT. Details in the Supporting Information. The resulting output file is provided on zenodo.

Reviewer #3 (Remarks to the Author)

-Line 2: replace "Fe4S4-OH2 and Fe8S9 clusters" with "[4Fe-4S]-OH2 and [8Fe-9S] clusters" for consistency throughout the text.

corrected as proposed

-Line 18: replace "Fe4S4 cluster" with "[4Fe-4S] cluster" for the consistency throughout the text.

corrected as proposed

-Line 132: Based on the data in Fig. S2b, the highest activity was at pH 8.8 rather than around pH 9.

-corrected

-Line 151-153: "The channel is about 20 Å long and at its bottom" is not clear. Where is the bottom? Based on Figure 2b, the 3'-phosphoadenosine and diphosphate moieties are not inside the channel, but in a pocket on the surface. It would be clearer if the figure were in a stereo view or if the channel lengths of BzCoA and water molecules were shown from the surface of the protein.

For clarification we showed the channel from two directions. The bottom is below the benzoyl head of the substrate. Yes, the 3'-phosphoadenosine and diphosphate moieties are not inside the channel, which can be visualized in figure 2b+c.

-Line 154: describe what kind of electron density is being displayed? AND replace "green" with "green mesh".

We wrote: The contour level of the $2F_{obs}-F_{calc}$ electron density (green mesh) for BzCoA is 1.6σ

-Line 154: replace "marked by" with "shown with".

changed as proposed

-Line 176: The authors have not provided any experimental evidence for the presence of a double cubane cluster (DCC) other than metal content, such as an omitted electron density together with an anomalous map if possible. It would be ideal to show the electron density for DCC, for example in Fig. 3a, but also in SI. Since the occupancy of the metal clusters can be well calculated at this resolution, the occupancy of the DCC and the active site clusters should be mentioned.

We have produced a new figure S3 that includes an anomalous and omit map for [8Fe-9S] cluster and [4Fe-4S]-OH₂ cluster. Both clusters are completely occupied. The FeS clusters and the ligating cysteine sulfurs are nearly on the same electron density level

-Line 181: although the anionic chloride ion did not fit the mut-S position, what about the lighter atoms such as oxygen?

-Lighter atoms can be excluded at this high resolution (see Figure S3)

-Line 191: in Fig. 3a indicate which sub-cluster is proximal or distal in the figure.

now marked in the figure

-Line 195: similar to the comments above for DCC, only the model is shown in the figure. Please include electron density maps, at least an omitted map.

Anomalous and omit maps are shown in the new figure S3.

-Line 205: what could be the role of the water chain, how does it interact with the protein, and is this chain still present in the structure of partially occupied CoA?

We don't know the function of the water chain. Perhaps the water chain does not play any specific role. In high-resolution electron density maps many waters can be identified and water chains are not unusual. The water chain links the [4Fe-4S] cluster with the border of the subunit and thereby interacts with residues Glu381, Asp328, Phe326, Ser130 and Arg378. In the structure of the partially occupied CoA the water chain is also visible.

-Line 207-210: the authors claim that the ligand at the open coordination site of the [4Fe-4S] cluster could be a water or hydroxy ligand, which actually makes sense for the mechanism proposed later, and -SH and Cl⁻ are ruled out because of the high resolution. However, if you play with occupancy for refinement, sulfur or Cl could fit with low occupancy and b-factor. Please comment on this.

The distance between the Fe-O is ca. 1.9 ± 0.1 Å based on the high-resolution electron density. The Fe-S (Cl) distance is ca. 2.2 - 2.3 Å. A distinction between O and lower occupied S is clearly possible.

-Line 212: draw or indicate "two polar interactions" in Fig. 3b.

We marked the two polar interactions (Glu381-N, Gly382-N) to the [4Fe-4S] clusters in Fig.3b

-Line 220: 1) there is no indication of the backbone nitrogen of Glu65 in the figure; 2) the solvent next to Gln124 and Glu65 is not H-bonded to Fe-OH₂/OH and Ser44, where the solvent is actually H-bonded to Gln124 and Glu65, and Ser44 is H-bonded to Glu65.

1) The backbone nitrogen of Glu65 is partly covered up but it is still to see. 2) The interpretation of reviewer is correct.

-Line 222: 1) Thr125 is not shown in the figure; 2) replace "-ND2" with "His131-ND2".

Thr125 was integrated and "-ND2" is replaced by "His131-ND2"

-Line 238: Tyr95 is not shown in Fig. 3b and Fig. S3. Instead there is Tyr97. It seems that Tyr97 is correct because Cys95 is there. Please check this.

Tyr97 is correct

-Line 240: It would be better to draw the carbon numbers of the benzoyl ring in Fig. 3b.

We marked atom C3 and C4 of the benzoyl ring

-Line 243: replace "Tyr95" with "Tyr97".

We corrected Tyr95 to Tyr97

-Line 248: Indicate C4 in benzoyl ring in Fig. S4, making the authors' claim more visible.

We marked atom C4 of the dienoyl ring

-Line 251: the authors claim that the crystal structure of BzdNO, free of substrate/product, shows a weakly occupied CoA ligand. The protein was heterologously produced by *E. coli*, purified and crystallized without substrate/product. The reviewer is interested to know where the CoA comes from, because *E. coli* would not produce large amounts of CoA? Second, the crystallization solution contains PEG molecules. If a molecule is partially occupied along with a translational pseudosymmetry character of the data set and diameter of the CoA channel, a short chain of PEG molecules may also be present in the channel. Please comment on this.

The reviewer is right that CoA could most likely derived from E. coli. The cellular CoA concentrations are in the μM range which should be sufficient to explain the partial occupancy.

It was mentioned in the text that the electron density is poor and the occupancy of the bound molecule is low. However without claiming a high reliability, the electron density shows a certain correlation with CoA. We write the sentence now with greater caution:

"A structural characterization of BzdNO without previous incubation with substrate/product does not reveal a completely empty substrate binding site but appears to contain besides several solvent molecules a weakly occupied CoA ligand."

-Line 255: large conformational change of residues 279-280 including very bulky residues (F, W, L) are moved into the CoA ester binding site to optimize the binding (Fig. S5). Is there any other changes in the structure due to this conformational change?

Only the conformational changes of residues 277-280 significantly affect the main chain. All other conformational changes are only related to minor side chain rearrangements (E411, E416, R75).

-Line 361: It appears that Tyr95 does not exist and Tyr97 may be correct. Please check the text.

We have corrected this error

-Line 455: Although the sequence similarity is low, the structure of the BzdO subunit is very similar to BzdN, a subunit of DCCP, and also to the HAD subunit with [4Fe-5S] cluster (beta subunit). It would be helpful to understand what has changed in the space of a sub-cluster during evolution by showing a superposition of these structures in SI.

In a new supplementary figure S10 we showed a superposition of BzdNO, HAD, DCCP and TudS (4-thiouridine-5'-monophosphate desulfidase).

-Line 571: As shown in Table S1, the Rwork/Rfree values after refinements are exceptionally high compared to the resolutions. This could be the presence of translational pseudosymmetry as described by the authors. Therefore, it would be better to explain this relationship in more detail in the Methods and report the degree of translational pseudosymmetry for each data set in Table S1.

We extended a sentence in the Method section and added the Patterson peak height relative to the origin indicating weak translational pseudosymmetry to Table S1.

Supporting Information

-Line 5: replace “Fe4S4-OH2 and Fe8S9 clusters” with “[4Fe-4S]-OH2 and [8Fe-9S] clusters” for the consistency throughout the text.

The names of the iron-sulfur clusters were made consistent.

-Line 21: incorporate PDB-ID of each data set in table S1.

We incorporated the PDB-IDs to table 1.

-Line 66: Black dotted lines for hydrophobic contacts are not shown in the figure. One-letter code for amino acids causes some confusion with atom names in the figure. Please use the three-letter code.

We omitted the hydrophobic interactions for clarity and correspondingly changes the legend of supplementary Fig. S3.

-Line 73: 1) Describe what kind of electron density is shown; 2) In a, mark C4 to show the out of the benzyl ring plane.

We used the $2F_{obs}-F_{calc}$ electron density. As proposed C4 is marked.

-Line 83: The partially CoA-free state is not colored in orange, but some light green with outlined backbone is there. clarify this.

The colors in the figure correspond with the description of the legend.

Line 110: Explanation of green and orange meshes is required.

Done

Point to point response to reviewer comments

Reviewer #2:

The manuscript has been revised significantly according to all reviewers' comments, and it could be published after addressing one minor concern:

1. The calculated relative energies of different spin states and spin populations on each metal ion should be listed as tables in the supporting information for friendly checking. It is very difficult to check the original files on Zenodo as the files are too large to be downloaded (more than 1GB).

Response:

We have listed the data as tables in the supplementary Information file as requested.